# Deep Generative Markov State Models

**Hao Wu**[1,2,*]**, Andreas Mardt**[1,*]**, Luca Pasquali**[1,*]**, and Frank Noe**[1,†]
[1]Dept. of Mathematics and Computer Science, Freie Universität Berlin, 14195 Berlin, Germany
[2]School of Mathematical Sciences, Tongji University, Shanghai, 200092, P.R. China

## Abstract

We propose a deep generative Markov State Model (DeepGenMSM) learning framework for inference of metastable dynamical systems and prediction of trajectories. After unsupervised training on time series data, the model contains (i) a probabilistic encoder that maps from high-dimensional configuration space to a small-sized vector indicating the membership to metastable (long-lived) states, (ii) a Markov chain that governs the transitions between metastable states and facilitates analysis of the long-time dynamics, and (iii) a generative part that samples the conditional distribution of configurations in the next time step. The model can be operated in a recursive fashion to generate trajectories to predict the system evolution from a defined starting state and propose new configurations. The DeepGenMSM is demonstrated to provide accurate estimates of the long-time kinetics and generate valid distributions for molecular dynamics (MD) benchmark systems. Remarkably, we show that DeepGenMSMs are able to make long timesteps in molecular configuration space and generate physically realistic structures in regions that were not seen in training data.

## 1 Introduction

Complex dynamical systems that exhibit events on vastly different timescales are ubiquitous in science and engineering. For example, molecular dynamics (MD) of biomolecules involve fast vibrations on the timescales of $10^{-15}$ seconds, while their biological function is often related to the rare switching events between long-lived states on timescales of $10^{-3}$ seconds or longer. In weather and climate systems, local fluctuations in temperature and pressure fields occur within minutes or hours, while global changes are often subject to periodic motion and drift over years or decades. Primary goals in the analysis of complex dynamical systems include:

1. *Deriving an interpretable model* of the essential long-time dynamical properties of these systems, such as the stationary behavior or lifetimes/cycle times of slow processes.

2. *Simulating the dynamical system*, e.g., to predict the system's future evolution or to sample previously unobserved system configurations.

A state-of-the-art approach for the first goal is to learn a Markovian model from time-series data, which is theoretically justified by the fact that physical systems are inherently Markovian. In practice, the long-time behavior of dynamical systems can be accurately described in a Markovian model when suitable features or variables are used, and when the time resolution of the model is sufficiently coarse such that the time-evolution can be represented with a manageable number of dynamical modes [24, 11]. In stochastic dynamical systems, such as MD simulation, variants of Markov state models (MSMs) are commonly used [3, 25, 22]. In MSMs, the configuration space is discretized, e.g., using a clustering method, and the dynamics between clusters are then described by a matrix

---

[*]H. Wu, A. Mardt and L. Pasquali equally contributed to this work.
[†]Author to whom correspondence should be addressed. Electronic mail: frank.noe@fu-berlin.de.

of transition probabilities [22]. The analogous approach for deterministic dynamical systems such as complex fluid flows is called Koopman analysis, where time propagation is approximated by a linear model in a suitable function space transformation of the flow variables [16, 26, 29, 4]. The recently proposed VAMPnets learn an optimal feature transformation from full configuration space to a low-dimensional latent space in which the Markovian model is built by variational optimization of a neural network [15]. When the VAMPnet has a probabilistic output (e.g. SoftMax layer), the Markovian model conserves probability, but is not guaranteed to be a valid transition probability matrix with nonnegative elements. A related work for deterministic dynamical systems is Extended Dynamic Mode Decomposition with dictionary learning [13]. All of these methods are purely analytic, i.e. they learn a reduced model of the dynamical system underlying the observed time series, but they miss a generative part that could be used to sample new time series in the high-dimensional configuration space.

Recently, several learning frameworks for dynamical systems have been proposed that partially address the second goal by including a decoder from the latent space back to the space of input features. Most of these methods primarily aim at obtaining a low-dimensional latent space that encodes the long-time behavior of the system, and the decoder takes the role of defining or regularizing the learning problem [30, 8, 14, 19, 23]. In particular none of these models have demonstrated the ability to generate viable structures in the high-dimensional configuration space, such as a molecular structure with realistic atom positions in 3D. Finally, some of these models learn a linear model of the long-timescale dynamics [14, 19], but none of them provide a probabilistic dynamical model that can be employed in a Bayesian framework. Learning the correct long-time dynamical behavior with a generative dynamical model is difficult, as demonstrated in [8].

Here, we address these aforementioned gaps by providing a deep learning framework that learns, based on time-series data, the following components:

1. Probabilistic encodings of the input configuration to a low-dimensional latent space by neural networks, $x_t \to \boldsymbol{\chi}(x_t)$.

2. A true transition probability matrix $\mathbf{K}$ describing the system dynamics in latent space for a fixed time-lag $\tau$:
$$\mathbb{E}\left[\boldsymbol{\chi}(x_{t+\tau})\right] = \mathbb{E}\left[\mathbf{K}^\top(\tau)\boldsymbol{\chi}(x_t)\right].$$
The probabilistic nature of the method allows us to train it with likelihood maximization and embed it into a Bayesian framework. In our benchmarks, the transition probability matrix approximates the long-time behavior of the underlying dynamical system with high accuracy.

3. A generative model from latent vectors back to configurations, allowing us to sample the transition density $\mathbb{P}(x_{t+\tau}|x_t)$ and thus propagate the model in configuration space. We show for the first time that this allows us to sample genuinely new and valid molecular structures that have not been included in the training data. This makes the method promising for performing active learning in MD [2, 21], and to predict the future evolution of the system in other contexts.

## 2 Deep Generative Markov State Models

Given two configurations $x, y \in \mathbb{R}^d$, where $\mathbb{R}^d$ is a potentially high-dimensional space of system configurations (e.g. the positions of atoms in a molecular system), Markovian dynamics are defined by the transition density $\mathbb{P}(x_{t+\tau} = y|x_t = x)$. Here we represent the transition density between $m$ states in the following form (Fig. 1):

$$\mathbb{P}(x_{t+\tau} = y|x_t = x) = \boldsymbol{\chi}(x)^\top \mathbf{q}(y;\tau) = \sum_{i=1}^m \chi_i(x) q_i(y;\tau). \tag{1}$$

Here, $\boldsymbol{\chi}(x)^\top = [\chi_1(x), ..., \chi_m(x)]$ represent the probability of configuration $x$ to be in a metastable (long-lived) state $i$

$$\chi_i(x) = \mathbb{P}(x_t \in \text{state } i \mid x_t = x).$$

Consequently, these functions are nonnegative ($\chi_i(x) \geq 0 \forall x$) and sum up to one ($\sum_i \chi_i(x) = 1 \forall x$). The functions $\boldsymbol{\chi}(x)$ can, e.g., be represented by a neural network mapping from $\mathbb{R}^d$ to $\mathbb{R}^m$ with a

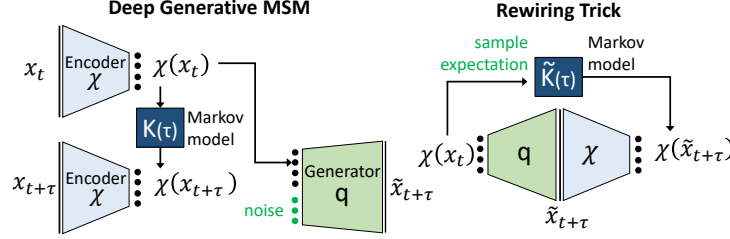

Figure 1: Schematic of Deep Generative Markov State Models (DeepGenMSMs) and the rewiring trick. The function $\chi$, here represented by neural networks, maps the time-lagged input configurations to metastable states whose dynamics are governed by a transition probability matrix $\mathbf{K}$. The generator samples the distribution $x_{t+\tau} \sim \mathbf{q}$ by employing a generative network that can produce novel configurations (or by resampling $x_{t+\tau}$ in DeepResampleMSMs). The rewiring trick consists of reconnecting the probabilistic networks $\mathbf{q}$ and $\chi$ such that the time propagation in latent space can be sampled: From the latent state $\chi(x_t)$, we generate a time-lagged configuration $x_{t+\tau}$ using $\mathbf{q}$, and then transform it back to the latent space, $\chi(x_{t+\tau})$. Each application of the rewired network samples the latent space transitions, thus providing the statistics to estimate the Markov model transition matrix $\mathbf{K}(\tau)$, which is needed for analysis. This trick allows $\mathbf{K}(\tau)$ to be estimated with desired constraints, such as detailed balance.

SoftMax output layer. Additionally, we have the probability densities

$$q_i(y;\tau) = \mathbb{P}(x_{t+\tau} = y | x_t \in \text{state } i)$$

that define the probability density of the system to "land" at configuration $y$ after making one time-step. We thus briefly call them "landing densities".

## 2.1 Kinetics

Before addressing how to estimate $\chi$ and $\mathbf{q}$ from data, we describe how to perform the standard calculations and analyses that are common in the Markov modeling field for a model of the form (1).

In Markov modeling, one is typically interested in the kinetics of the system, i.e. the long-time behavior of the dynamics. This is captured by the elements of the transition matrix $\mathbf{K} = [k_{ij}]$ between metastable states. $\mathbf{K}$ can be computed as follows: the product of the probability density to jump from metastable $i$ to a configuration $y$ and the probability that this configuration belongs to metastable state $j$, integrated over the whole configuration space.

$$k_{ij}(\tau) = \int_y q_i(y;\tau)\chi_j(y)\,\mathrm{d}y. \tag{2}$$

Practically, this calculation is implemented via the "rewiring trick" shown in Fig. 1, where the configuration space integral is approximated by drawing samples from the generator. The estimated probabilistic functions $\mathbf{q}$ and $\chi$ define, by construction, a valid transition probability matrix $\mathbf{K}$, i.e. $k_{ij} \geq 0$ and $\sum_j k_{ij} = 1$. As a result, the proposed models have a structural advantage over other high-accuracy Markov state modeling approaches that define metastable states in a fuzzy or probabilistic manner but do not guarantee a valid transition matrix [12, 15] (See Supplementary Material for more details.).

The stationary (equilibrium) probabilities of the metastable states are given by the vector $\boldsymbol{\pi} = [\pi_i]$ that solves the eigenvalue problem with eigenvalue $\lambda_1 = 1$:

$$\boldsymbol{\pi} = \mathbf{K}^\top \boldsymbol{\pi}, \tag{3}$$

and the stationary (equilibrium) distribution in configuration space is given by:

$$\mu(y) = \sum_i \pi_i q_i(y;\tau) = \boldsymbol{\pi}^\top \mathbf{q}(y;\tau). \tag{4}$$

Finally, for a fixed definition of states *via* $\chi$, the self-consistency of Markov models may be tested using the Chapman-Kolmogorov equation

$$\mathbf{K}^n(\tau) \approx \mathbf{K}(n\tau) \tag{5}$$

which involves estimating the functions $\mathbf{q}(y; n\tau)$ at different lag times $n\tau$ and comparing the resulting transition matrices with the $n$th power of the transition matrix obtained at lag time $\tau$. A consequence of Eq. (5) is that the relaxation times

$$t_i(\tau) = -\frac{\tau}{\log|\lambda_i(\tau)|} \tag{6}$$

are independent of the lag time $\tau$ at which $\mathbf{K}$ is estimated [27]. Here, $\lambda_i$ with $i = 2, ..., m$ are the nontrivial eigenvalues of $\mathbf{K}$.

## 2.2 Maximum Likelihood (ML) learning of DeepResampleMSM

Given trajectories $\{x_t\}_{t=1,...,T}$, how do we estimate the membership probabilities $\boldsymbol{\chi}(x)$, and how do we learn and sample the landing densities $\mathbf{q}(y)$? We start with a model, where $\mathbf{q}(y)$ are directly derived from the observed (empirical) observations, i.e. they are point densities on the input configurations $\{x_t\}$, given by:

$$q_i(y) = \frac{1}{\bar{\gamma}_i}\gamma_i(y)\rho(y). \tag{7}$$

Here, $\rho(y)$ is the empirical distribution, which in the case of finite sample size is simply $\rho(y) = \frac{1}{T-\tau}\sum_{t=1}^{T-\tau}\delta(y - x_{t+\tau})$, and $\gamma_i(y)$ is a trainable weighting function. The normalization factor $\bar{\gamma}_i = \frac{1}{T-\tau}\sum_{t=1}^{T-\tau}\gamma_i(x_{t+\tau}) = \mathbb{E}_{y\sim\rho_1}[\gamma_i(y)]$ ensures $\int_y q_i(y)\,\mathrm{d}y = 1$.

Now we can optimize $\chi_i$ and $\gamma_i$ by maximizing the likelihood (ML) of generating the pairs $(x_t, x_{t+\tau})$ observed in the data. The log-likelihood is given by:

$$LL = \sum_{t=1}^{T-\tau}\ln\left(\sum_{i=1}^{m}\chi_i(x_t)\bar{\gamma}_i^{-1}\gamma_i(x_{t+\tau})\right), \tag{8}$$

and is maximized to train a deep MSM with the structure shown in Fig. 1.

Alternatively, we can optimize $\chi_i$ and $\gamma_i$ using the Variational Approach for Markov Processes (VAMP) [31]. However, we found the ML approach to perform significantly better in our tests, and we thus include the VAMP training approach only in the Supplementary Material without elaborating on it further.

Given the networks $\boldsymbol{\chi}$ and $\boldsymbol{\gamma}$, we compute $\mathbf{q}$ from Eq. (7). Employing the rewiring trick shown in Fig. 1 results in computing the transition matrix by a simple average over all configurations:

$$\mathbf{K} = \frac{1}{N}\sum_{t=\tau}^{T-\tau}\mathbf{q}(x_{t+\tau})\boldsymbol{\chi}(x_{t+\tau})^\top. \tag{9}$$

The deep MSMs described in this section are neural network generalizations of traditional MSMs – they learn a mapping from configurations to metastable states, where they aim obtaining a good approximation of the kinetics of the underlying dynamical system, by means of the transition matrix $\mathbf{K}$. However, since the landing distribution $\mathbf{q}$ in these methods is derived from the empirical distribution (7), any generated trajectory will only resample configurations from the input data. To highlight this property, we will refer to the deep MSMs with the present methodology as DeepResampleMSM.

## 2.3 Energy Distance learning of DeepGenMSM

In contrast to DeepResampleMSM, we now want to learn deep generative MSM (DeepGenMSM), which can be used to generate trajectories that do not only resample from input data, but can produce genuinely new configurations. To this end, we train a generative model to mimic the empirical distribution $q_i(y)$:

$$y = G(e_i, \epsilon), \tag{10}$$

where the vector $e_i \in \mathbb{R}^m$ is a one-hot encoding of the metastable state, and $\epsilon$ is a i.i.d. random vector where each component samples from a Gaussian normal distribution.

Here we train the generator $G$ by minimizing the conditional Energy Distance (ED), whose choice is motivated in the Supplementary Material. The standard ED, introduced in [28], is a metric between the distributions of random vectors, defined as

$$D_E\left(\mathbb{P}(\mathbf{x}), \mathbb{P}(\mathbf{y})\right) = \mathbb{E}\left[2\|x - y\| - \|x - x'\| - \|y - y'\|\right] \tag{11}$$

for two real-valued random variables $\mathbf{x}$ and $\mathbf{y}$. $x, x', y, y'$ are independently distributed according to the distributions of $\mathbf{x}, \mathbf{y}$. Based on this metric, we introduce the conditional energy distance between the transition density of the system and that of the generative model:

$$
\begin{aligned}
D &\triangleq \mathbb{E}\left[D_E\left(\mathbb{P}(\mathbf{x}_{t+\tau}|x_t), \mathbb{P}(\hat{\mathbf{x}}_{t+\tau}|x_t)\right)|x_t\right] \\
&= \mathbb{E}\left[2\left\|\hat{x}_{t+\tau} - x_{t+\tau}\right\| - \left\|\hat{x}_{t+\tau} - \hat{x}'_{t+\tau}\right\| - \left\|x_{t+\tau} - x'_{t+\tau}\right\|\right]
\end{aligned}
\tag{12}
$$

Here $x_{t+\tau}$ and $x'_{t+\tau}$ are distributed according to the transition density for given $x_t$ and $\hat{x}_{t+\tau}, \hat{x}'_{t+\tau}$ are independent outputs of the generative model conditioned on $x_t$. Implementing the expectation value with an empirical average results in an estimate for $D$ that is unbiased, up to an additive constant. We train $G$ to minimize $D$. See Supplementary Material for detailed derivations and the training algorithm used.

After training, the transition matrix can be obtained by using the rewiring trick (Fig. 1), where the configuration space integral is sampled by generating samples from the generator:

$$
[\mathbf{K}]_{ij} = \mathbb{E}_\epsilon\left[\chi_j\left(G(e_i, \epsilon)\right)\right].
\tag{13}
$$

## 3 Results

Below we establish our framework by applying it to two well-defined benchmark systems that exhibit metastable stochastic dynamics. We validate the stationary distribution and kinetics by computing $\boldsymbol{\chi}(x)$, $\mathbf{q}(y)$, the stationary distribution $\mu(y)$ and the relaxation times $t_i(\tau)$ and comparing them with reference solutions. We will also test the abilities of DeepGenMSMs to generate physically valid molecular configurations.

The networks were implemented using PyTorch [20] and Tensorflow [6]. For the full code and all details about the neural network architecture, hyper-parameters and training algorithm, please refer to https://github.com/markovmodel/deep_gen_msm.

### 3.1 Diffusion in Prinz potential

We first apply our framework to the time-discretized diffusion process $x_{t+\Delta t} = -\Delta t \nabla V(x_t) + \sqrt{2\Delta t}\dot{\eta}_t$ with $\Delta t = 0.01$ in the Prinz potential $V(x_t)$ introduced in [22] (Fig. 2a). For this system we know exact results for benchmarking: the stationary distribution and relaxation timescales (black lines in Fig. 2b,c) and the transition density (Fig. 2d). We simulate trajectories of lengths $250,000$ and $125,000$ time steps for training and validation, respectively. For all methods, we repeat the data generation and model estimation process 10 times and compute mean and standard deviations for all quantities of interest, which thus represent the mean and variance of the estimators.

The functions $\chi$, $\gamma$ and $G$ are represented with densely connected neural networks. The details of the architecture and the training procedure can be found in the Supplementary Information.

We compare DeepResampleMSMs and DeepGenMSMs with standard MSMs using four or ten states obtained with $k$-means clustering. Note that standard MSMs do not directly operate on configuration space. When using an MSM, the transition density (Eq. 1) is thus simulated by:

$$
x_t \xrightarrow{\chi(x_t)} i \xrightarrow{\sim \mathbf{K}_{i,*}} j \xrightarrow{\sim \rho_j(y)} x_{t+\tau},
$$

i.e., we find the cluster $i$ associated with a configuration $x_t$, which is deterministic for regular MSMs, then sample the cluster $j$ at the next time-step, and sample from the conditional distribution of configurations in cluster $j$ to generate $x_{t+\tau}$.

Both DeepResampleMSMs trained with the ML method and standard MSMs can reproduce the stationary distribution within statistical uncertainty (Fig. 2b). For long lag times $\tau$, all methods converge from below to the correct relaxation timescales (Fig. 2c), as expected from theory [22, 18]. When using equally many states (here: four), the DeepResampleMSM has a much lower bias in the relaxation timescales than the standard MSM. This is expected from approximation theory, as the DeepResampleMSMs represents the four metastable states with a meaningful, smooth membership functions $\boldsymbol{\chi}(x_t)$, while the four-state MSM cuts the memberships hard at boundaries with low sample density (Supplementary Fig. 1). When increasing the number of metastable states, the bias of all estimators will reduce. An MSM with ten states is needed to perform approximately equal to a four-state DeepResampleMSM (Fig. 2c). All subsequent analyses use a lag time of $\tau = 5$.

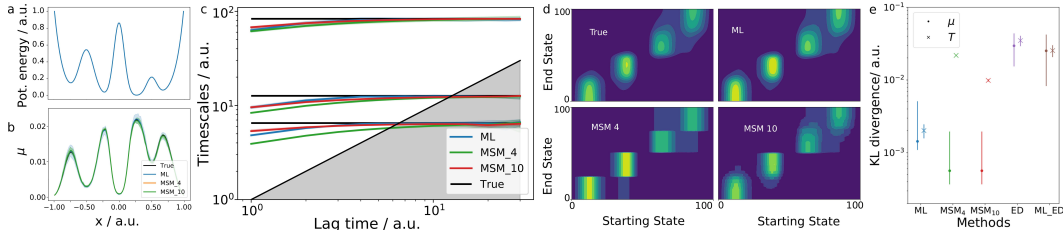

Figure 2: Performance of deep versus standard MSMs for diffusion in the Prinz Potential. (a) Potential energy as a function of position $x$. (b) Stationary distribution estimates of all methods with the exact distribution (black). (c) Implied timescales of the Prinz potential compared to the real ones (black line). (d) True transition density and approximations using maximum likelihood (ML) DeepResampleMSM, four and ten state MSMs. (e) KL-divergence of the stationary and transition distributions with respect to the true ones for all presented methods (also DeepGenMSM).

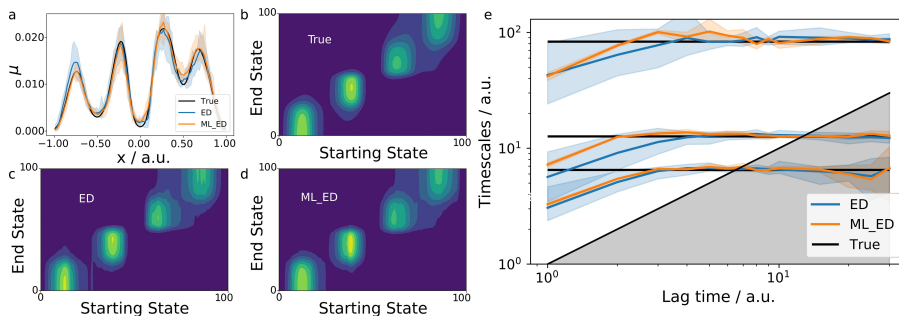

Figure 3: Performance of DeepGenMSMs for diffusion in the Prinz Potential. Comparison between exact reference (black), DeepGenMSMs estimated using only energy distance (ED) or combined ML-ED training. (a) Stationary distribution. (b-d) Transition densities. (e) Relaxation timescales.

The DeepResampleMSM generates a transition density that is very similar to the exact density, while the MSM transition densities are coarse-grained by virtue of the fact that $\boldsymbol{\chi}(x_t)$ performs a hard clustering in an MSM (Fig. 2d). This impression is confirmed when computing the Kullback-Leibler divergence of the distributions (Fig. 2e).

Encouraged by the accurate results of DeepResampleMSMs, we now train DeepGenMSM, either by training both the $\boldsymbol{\chi}$ and $\mathbf{q}$ networks by minimizing the energy distance (ED), or by taking $\boldsymbol{\chi}$ from a ML-trained DeepResampleMSM and only training the $\mathbf{q}$ network by minimizing the energy distance (ML-ED). The stationary densities, relaxation timescales and transition densities can still be approximated in these settings, although the DeepGenMSMs exhibit larger statistical fluctuations than the resampling MSMs (Fig. 3). ML-ED appears to perform slightly better than ED alone, likely because reusing $\boldsymbol{\chi}$ from the ML training makes the problem of training the generator easier.

For a one-dimensional example like the Prinz potential, learning a generative model does not provide any added value, as the distributions can be well approximated by the empirical distributions. The fact that we can still get approximately correct results for stationary, kinetics and dynamical properties encourages us to use DeepGenMSMs for a higher-dimensional example, where the generation of configurations is a hard problem.

### 3.2 Alanine dipeptide

We use explicit-solvent MD simulations of Alanine dipeptide as a second example. Our aim is the learn stationary and kinetic properties, but especially to learn a generative model that generates genuinely novel but physically meaningful configurations. One $250\,\mathrm{ns}$ trajectory with a storage interval of $1\,\mathrm{ps}$ is used and split $80\%/20\%$ for training and validation – see [15] for details of the simulation setup. We characterize all structures by the three-dimensional Cartesian coordinates of the heavy atoms, resulting in a 30 dimensional configuration space. While we do not have exact results for Alanine dipeptide, the system is small enough and well enough sampled, such that high-

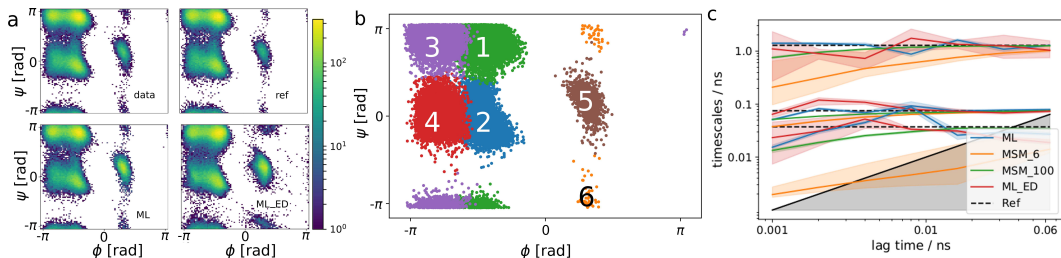

Figure 4: Performance of DeepResampleMSM and DeepGenMSMs versus standard MSMs on the Alanine dipeptide simulation trajectory. (a) Data distribution and stationary distributions from reference MSM, DeepResampleMSM, and DeepGenMSM. (b) State classification by DeepResampleMSM (c) Relaxation timescales.

quality estimates of stationary and kinetic properties can be obtained from a very fine MSM [22]. We therefore define an MSM build on 400 equally sized grid areas in the $(\phi, \psi)$-plane as a reference at a lag time of $\tau = 25\,\mathrm{ps}$ that has been validated by established methods [22].

Neural network and training details are again found at the git repository and in the Supplementary Information.

For comparison with deep MSMs, we build two standard MSMs following a state of the art protocol: we transform input configurations with a kinetic map preserving 95% of the cumulative kinetic variance [17], followed by $k$-means clustering, where $k = 6$ and $k = 100$ are used.

DeepResampleMSM trained with ML method approximate the stationary distribution very well (Fig. 4a). The reference MSM assigns a slightly lower weight to the lowest-populated state 6, but otherwise the data, reference distribution and deep MSM distribution are visually indistinguishable. The relaxation timescales estimated by a six-state DeepResampleMSM are significantly better than with six-state standard MSMs. MSMs with 100 states have a similar performance as the deep MSMs but this comes at the cost of a model with a much larger latent space.

Finally, we test DeepGenMSMs for Alanine dipeptide where $\chi$ is trained with the ML method and the generator is then trained using ED (ML-ED). The stationary distribution generated by simulating the DeepGenMSM recursively results in a stationary distribution which is very similar to the reference distribution in states 1-4 with small $\phi$ values (Fig. 4a). States number 5 and 6 with large $\phi$ values are captured, but their shapes and weights are somewhat distorted (Fig. 4a). The one-step transition densities predicted by the generator are high quality for all states (Suppl. Fig. 2), thus the differences observed for the stationary distribution must come from small errors made in the transitions between metastable states that are very rarely observed for states 5 and 6. These rare events result in poor training data for the generator. However, the DeepGenMSMs approximates the kinetics well within the uncertainty that is mostly due to estimator variance (Fig. 4c).

Now we ask whether DeepGenMSMs can sample valid structures in the 30-dimensional configuration space, i.e., if the placement of atoms is physically meaningful. As we generate configurations in Cartesian space, we first check if the internal coordinates are physically viable by comparing all bond lengths and angles between real MD data and generated trajectories (Fig. 5). The true bond lengths and angles are almost perfectly Gaussian distributed, and we thus normalize them by shifting each distribution to a mean of 0 and scaling it to have standard deviation 1, which results all reference distributions to collapse to a normal distribution (Fig. 5a,c). We normalize the generated distribution with the mean and standard distribution of the true data. Although there are clear differences (Fig. 5b,d), these distributions are very encouraging. Bonds and angles are very stiff degrees of freedom, and the fact that most differences in mean and standard deviation are small when compared to the true fluctuation width means that the generated structures are close to physically accurate and could be refined by little additional MD simulation effort.

Finally, we perform an experiment to test whether the DeepGenMSM is able to generate genuinely new configurations that do exist for Alanine dipeptide but have not been seen in the training data. In other words, can the generator "extrapolate" in a meaningful way? This is a fundamental question, because simulating MD is exorbitantly expensive, with each simulation time step being computationally expensive but progressing time only of the order of $10^{-15}$ seconds, while often total simulation

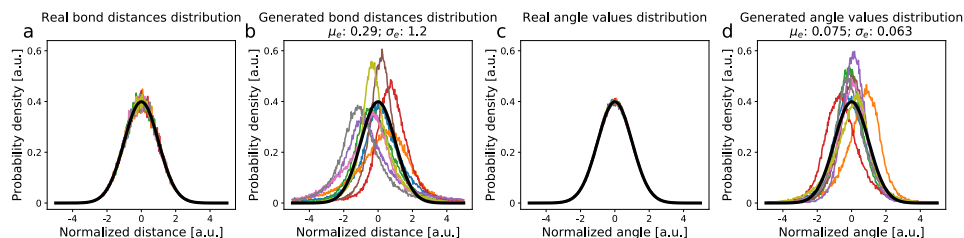

Figure 5: Normalized bond (a,b) and angle (c,d) distributions of Alanine dipeptide compared to Gaussian normal distribution (black). (a,c) True MD data. (b,d) Data from trajectories generated by DeepGenMSMs.

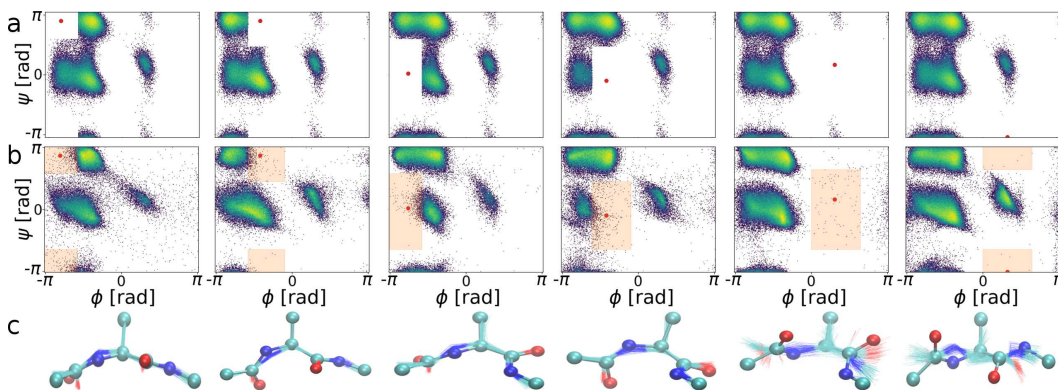

Figure 6: DeepGenMSMs can generate physically realistic structures in areas that were not included in the training data. (a) Distribution of training data. (b) Generated stationary distribution. (c) Representative "real" molecular configuration (from MD simulation) in each of the metastable states (sticks and balls), and the 100 closest configurations generated by the DeepGenMSM (lines).

timescales of $10^{-3}$ seconds or longer are needed. A DeepGenMSM that makes leaps of length $\tau$ – orders of magnitude larger than the MD simulation time-step – and has even a small chance of generating new and meaningful structures would be extremely valuable to discover new states and thereby accelerate MD sampling.

To test this ability, we conduct six experiments, in each of which we remove all data belonging to one of the six metastable states of Alanine dipeptide (6a). We train a DeepGenMSM with each of these datasets separately, and simulate it to predict the stationary distribution (6b). While the generated stationary distributions are skewed and the shape of the distribution in the $(\phi, \psi)$ range with missing-data are not quantitatively predicted, the DeepGenMSMs do indeed predict configurations where no training data was present (6b). Surprisingly, the quality of most of these configurations is high (6c). While the structures of the two low-populated states 5-6 do not look realistic, each of the metastable states 1-4 are generated with high quality, as shown by the overlap of a real MD structure and the 100 most similar generated structures (6c).

In conclusion, deep MSMs provide high-quality models of the stationary and kinetic properties for stochastic dynamical systems such as MD simulations. In contrast to other high-quality models such as VAMPnets, the resulting model is truly probabilistic and can thus be physically interpreted and be used in a Bayesian framework. For the first time, it was shown that generating dynamical trajectories in a 30-dimensional molecular configuration space results in sampling of physically realistic molecular structures. While Alanine dipeptide is a small system compared to proteins and other macromolecules that are of biological interest, our results demonstrate that efficient sampling of new molecular structures is possible with generative dynamic models, and improved methods can be built upon this. Future methods will especially need to address the difficulties of generating valid configurations in low-probability regimes, and it is likely that the energy distance used here for generator training needs to be revisited to achieve this goal.

**Acknowledgements** This work was funded by the European Research Commission (ERC CoG "ScaleCell"), Deutsche Forschungsgemeinschaft (CRC 1114/A04, Transregio 186/A12, NO 825/4–1, Dynlon P8), and the "1000-Talent Program of Young Scientists in China".

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
