[Supplementary Material · summary_for_discussion_revision_final_supplement.pdf]

## Supplementary Material

**Analysis of transition matrices**

The transition matrix $\mathbf{K}(\tau) = [k_{ij}(\tau)] \in \mathbb{R}^{m \times m}$ is defined as

$$k_{ij}(\tau) = \mathbb{P}(x_{t+\tau} \in \text{state } j | x_t \in \text{state } i).$$

Then, according to (1), we have

$$
\begin{aligned}
k_{ij}(\tau) &= \int \mathbb{P}(x_{t+\tau} = y | x_t \in \text{state } i) \cdot \mathbb{P}(x_{t+\tau} \in \text{state } j | x_{t+\tau} = y) \mathrm{d}y \\
&= \int q_i(y; \tau) \chi_j(y) \, \mathrm{d}y,
\end{aligned}
$$

and

$$k_{ij}(n\tau) = [\mathbf{K}(\tau)^n]_{ij}.$$

If $\chi, \mathbf{q}$ satisfy the conditions

$$
\begin{aligned}
&\chi_i(x) \geq 0, \sum_j \chi_j(x) = 1, \\
&q_i(x; \tau) \geq 0, \int q_i(y; \tau) \mathrm{d}y = 1, \quad \forall x, i,
\end{aligned}
$$

we have $k_{ij}(\tau) \geq 0$ and $\sum_j k_{ij}(\tau) = 1$, i.e., $\mathbf{K}(\tau)$ computed from $\chi, \mathbf{q}$ is a valid transition probability matrix.

For the distribution $\mu$ defined in (4),

$$
\begin{aligned}
\int \mathbb{P}(x_{t+\tau} = y | x_t = x) \cdot \mu(x) \mathrm{d}x &= \int \mathbf{q}(y; \tau)^\top \chi(x) \cdot \mathbf{q}(x; \tau)^\top \pi \mathrm{d}x \\
&= \mathbf{q}(y; \tau)^\top \pi \\
&= \mu(y),
\end{aligned}
$$

which shows $\mu$ is the stationary distribution of model (1).

**VAMP-E training of deep MSMs**

An alternative to ML training is to employ a score from the Variational Approach of Markov Processes (VAMP) [31]. The VAMP-E score has the advantage over other VAMP scores employed previously [15] that we do not have to specify the rank of the model [31]. The VAMP-E score is computed as

$$\mathcal{R}_E = \mathrm{tr}\left( 2\mathbf{C}_{01}\bar{\mathbf{\Gamma}}^{-1} - \mathbf{C}_{00}\bar{\mathbf{\Gamma}}^{-1}\mathbf{C}_{11}\bar{\mathbf{\Gamma}}^{-1} \right) \tag{14}$$

which depends on covariance matrices estimated from the transformed data:

$$
\begin{aligned}
[\mathbf{C}_{00}]_{ij} &= \mathbb{E}_t[\chi_i(x_t)\chi_j(x_t)] \\
[\mathbf{C}_{11}]_{ij} &= \mathbb{E}_t[\gamma_i(x_{t+\tau})\gamma_j(x_{t+\tau})] \\
[\mathbf{C}_{01}]_{ij} &= \mathbb{E}_t[\chi_i(x_t)\gamma_j(x_{t+\tau})] \\
\bar{\mathbf{\Gamma}} &= \mathrm{diag}(\bar{\gamma}_1, \ldots, \bar{\gamma}_m).
\end{aligned}
$$

We can use the standard empirical estimators to compute $\mathbb{E}_t$. We can then train a deep MSM using the structure shown in Fig. 1 by maximizing (14).

**Using the Energy Distance to train generative networks**

The Energy Distance (ED) [28] is a metric that measures the difference between the distributions of two real valued random vectors $x$ and $y$, and is defined as

$$D_E(\mathbb{P}(x), \mathbb{P}(y)) = \mathbb{E}\left[2\|x - y\| - \|x - x'\| - \|y - y'\|\right]. \tag{15}$$

Here, $x', y'$ are independently distributed according to the distributions of $y, z$. Therefore, the conditional energy distance given in (12) is equal to the mean value of the energy distance between the

conditional distributions $\mathbb{P}(x_{t+\tau}|x_t)$ and $\mathbb{P}(\hat{x}_{t+\tau}|x_t)$ for all $x_t$, and satisfies that $D \geq 0$ and $D = 0$ if and only if $\mathbb{P}(x_{t+\tau}|x_t) = \mathbb{P}(\hat{x}_{t+\tau}|x_t)$ for all $x_t$.

Noticing that $\mathbb{E}\left[\left\|x_{t+\tau} - x'_{t+\tau}\right\|\right]$ is a constant for a given system. We can therefore approximate $D$ as

$$
\begin{aligned}
D &= \mathbb{E}\left[\|\hat{x}_{t+\tau} - x_{t+\tau}\| + \|\hat{x}'_{t+\tau} - x_{t+\tau}\| - \|\hat{x}_{t+\tau} - \hat{x}'_{t+\tau}\|\right] + \text{const} \\
&= \mathbb{E}[d_t] + \text{const} \\
&\approx \frac{1}{N}\sum_t d_t + \text{const}
\end{aligned}
$$

where

$$
d_t = \|G(e_{I_t}, \epsilon_t) - x_{t+\tau}\| + \|G(e_{I'_t}, \epsilon'_t) - x_{t+\tau}\| - \|G(e_{I_t}, \epsilon_t) - G(e_{I'_t}, \epsilon'_t)\| \tag{16}
$$

Here, $N = T - \tau$ is the number of all transition pairs $(x_t, x_{t+\tau})$ present in the trajectory data, $I_t, I'_t$ are discrete random variables with $\mathbb{P}(I_t = i) = \mathbb{P}(I'_t = i) = \chi_i(x_t)$, and $\epsilon_t, \epsilon'_t$ are i.i.d random vectors whose components have Gaussian normal distributions.

The gradient of $D$ with respect to parameters $W_G$ of the generative model $G$ can be unbiasedly estimated by the mean value of $\partial d_t / \partial W_G$. But for parameters $W_\chi$ of $\chi$, $\partial d_t / \partial W_\chi$ does not exist because $I_t, I'_t$ is discrete-valued. In order to overcome this problem, we assume here $\chi(x) = \text{SoftMax}\left[\mathbf{o}(x)\right]$ is modeled by a neural network with the softmax output layer. Then

$$
\begin{aligned}
\frac{\partial}{\partial o_k}\mathbb{E}[d_t|x_t, x_{t+\tau}] &= \sum_{i,j}\chi_i(x)\chi_j(x)\left(1_{i=k} + 1_{j=k} - 2\chi_k(x_t)\right) \\
&\quad \cdot \mathbb{E}\left[\|G(e_i, \epsilon_t) - x_{t+\tau}\| + \|G(e_j, \epsilon'_t) - x_{t+\tau}\| - \|G(e_i, \epsilon_t) - G(e_j, \epsilon'_t)\|\right] \\
&= \mathbb{E}\left[\left(1_{I_t=k} + 1_{I'_t=k} - 2\chi_k(x_t)\right) \cdot d_t\right],
\end{aligned}
$$

which leads to the estimation

$$
\begin{aligned}
\frac{\partial D}{\partial W_\chi} &= \sum_k \frac{\partial o_k}{\partial W_\chi}\frac{\partial D}{\partial o_k} \\
&\approx \frac{1}{N}\sum_t d_t \sum_k \left(1_{I_t=k} + 1_{I'_t=k} - 2\chi_k(x_t)\right)\frac{\partial o_k}{\partial W_\chi}.
\end{aligned}
$$

By using the stochastic gradient over mini-batch over the entire data, we can train the generative MSM as follows the subsequent algorithm:

1. Randomly choose a mini-batch $\{(x_{(n)}, y_{(n)})\}_{i=1}^B \subset \{(x_t, x_{t+\tau})\}$ with batch size $B$.
2. Draw $I_{(n)}, I'_{(n)}$ with
$$
\mathbb{P}(I_{(n)} = i) = \mathbb{P}(I'_{(n)} = i) = \chi_i(\tilde{x}_{(n)}), \tag{17}
$$
   and draw $\epsilon_{(n)}, \epsilon'_{(n)}$ according to the Gaussian distribution for $n = 1, \ldots, B$.
3. Compute
$$
\begin{aligned}
\delta W_G &= \frac{1}{B}\sum_{n=1}^B \frac{\partial d_{(n)}}{\partial W_G} \\
\delta W_\chi &= \frac{1}{B}\sum_{i=1}^B d_{(n)} \cdot \sum_{k=1}^m \left(1_{I_{(n)}=k} + 1_{I'_{(n)}=k} - 2\chi_k(x_{(n)})\right)\frac{\partial o_k(x_{(n)})}{\partial W_\chi} \tag{18}
\end{aligned}
$$
   with
$$
d_{(n)} = \left\|G(e_{I_{(n)}}, \epsilon_{(n)}) - y_{(n)}\right\| + \left\|G(e_{I'_{(n)}}, \epsilon'_{(n)}) - y_{(n)}\right\| - \left\|G(e_{I_{(n)}}, \epsilon_{(n)}) - G(e_{I'_{(n)}}, \epsilon'_{(n)})\right\| \tag{19}
$$
   and $\chi = \text{SoftMax}\left[\mathbf{o}\right]$.
4. Update
$$
\begin{aligned}
W_G &\leftarrow W_G - \eta\delta W_G \\
W_P &\leftarrow W_P - \eta\delta W_P
\end{aligned}
$$
   with a learning rate $\eta$.

**Motivation of Energy Distance as the training metric**

The major advantages of ED are:

1. It can be unbiasedly estimated from the data without an extra "adversarial" network as in GANs.

2. Unlike the KL divergence (see example 1 in [1]), ED does not diverge in the case of few data points (small batch sizes) or low populated probability density areas.

3. As a specific Maximum Mean Discrepancy (MMD), ED can avoid the problem of popular kernel-MMDs that the gradients of cost functions are vanished if the generated samples are far away from the training data, and therefore achieve higher efficiency when learning generative models.

**Network architecture and training procedure**

All neural networks representing the functions $\chi$, $\gamma$ and $G$ for the Prinz potential are using 64 nodes in all 4 hidden layers and batch normalization after each layer [9]. Rectified linear activation functions (ReLUs) are used, except for the output layer of $\chi$ which uses SoftMax and the output layer of $G$ which has a linear activation function. Both $\chi$ and $\gamma$ have 4 output nodes, and $G$ receives a four-dimensional 1-hot-encoding of the metastable state plus a four-dimensional noise vector as inputs. Optimization is done using Adam [10], with early stopping checking if the validation score is not increasing over 5 epochs. The learning rate for the training of $\chi$, $\gamma$ is $\lambda = 10^{-3}$, and for $G$ $\lambda = 10^{-5}$ with a batchsize of 100. We are using a time-lag of $\tau = 5$ frames.

For alanine dipeptide, $\chi$ and $\gamma$ consist both of 3 residual blocks [7] built of 3 layers all having 100 nodes, with exponential linear units (ELUs) [5], and batch normalization for each layer. The output layer has 6 output nodes, where $\chi$ uses a softmax activation function and $\gamma$ a RELU, respectively. In order to find all slow processes, it was necessary to pre-train $\chi$ with the VAMPnet method [15]. The generator $G$ uses 6 noise inputs and a six-dimensional 1-hot-encoding of the metastable state and the ML-ED scheme. Networks are trained with Adam until the validation score converges with a learning rate of $\lambda = 10^{-5}$ for $\chi$, $\gamma$ using 8000 as batchsize and $\lambda = 10^{-4}$ for $G$ using 1500 frames for a batch. All subsequent analyses that use a fixed lag time employ $\tau = 1$ ps.

For finding the hyperparameter we performed a restricted grid search, which showed comparing the KL divergence between the modeled distributions that the result does only marginally depend on the choice of the parameters (see 1 for an example).

**Supplementary Figures**

Supplementary Fig. 1: $\chi(x)$ of the Prinz potential (a) Potential energy as a function of position x. (b) Maximum Likelihood (c) four state MSM (d) 10 state MSM (e) energy distance.

| depth | width | dim random | KL div. / $10^{-2}$ |
|---|---|---|---|
| 2 | 16 | 1 | 1.7 |
| 2 | 16 | 2 | 2.2 |
| 2 | 16 | 4 | 2.3 |
| 2 | 32 | 1 | 2.2 |
| 2 | 32 | 2 | 2.2 |
| 2 | 32 | 4 | 2.5 |
| 2 | 64 | 1 | 2.4 |
| 2 | 64 | 2 | 2.7 |
| 2 | 64 | 4 | 2.8 |
| 2 | 128 | 1 | 2.7 |
| 2 | 128 | 2 | 3.2 |
| 2 | 128 | 4 | 3.8 |
| 4 | 16 | 1 | 2.0 |
| 4 | 16 | 2 | 1.6 |
| 4 | 16 | 4 | 2.8 |
| 4 | 32 | 1 | 1.7 |
| 4 | 32 | 2 | 3.4 |
| 4 | 32 | 4 | 3.2 |
| 4 | 64 | 1 | 2.3 |
| 4 | 64 | 2 | 2.9 |
| 4 | 64 | 4 | 3.5 |
| 4 | 128 | 1 | 1.8 |
| 4 | 128 | 2 | 2.8 |
| 4 | 128 | 4 | 1.8 |
| 6 | 16 | 1 | 1.5 |
| 6 | 16 | 2 | 3.5 |
| 6 | 16 | 4 | 2.5 |
| 6 | 32 | 1 | 3.3 |
| 6 | 32 | 2 | 1.9 |
| 6 | 32 | 4 | 2.3 |
| 6 | 64 | 1 | 1.5 |
| 6 | 64 | 2 | 2.7 |
| 6 | 64 | 4 | 2.5 |
| 6 | 128 | 1 | 1.6 |
| 6 | 128 | 2 | 3.1 |
| 6 | 128 | 4 | 1.7 |
| 8 | 16 | 1 | 1.8 |
| 8 | 16 | 2 | 1.9 |
| 8 | 16 | 4 | 2.0 |
| 8 | 32 | 1 | 1.6 |
| 8 | 32 | 2 | 2.2 |
| 8 | 32 | 4 | 2.7 |
| 8 | 64 | 1 | 1.2 |
| 8 | 64 | 2 | 2.4 |
| 8 | 64 | 4 | 2.2 |
| 8 | 128 | 1 | 1.8 |
| 8 | 128 | 2 | 2.0 |
| 8 | 128 | 4 | 1.6 |

Table 1: Hyperparameter comparison of the KL divergence of the generated stationary distribution with respect to the true one for the Prinz potential varying the depth, the width, and the random input dimension taking the mean over 5 runs.

Supplementary Fig. 2: Conditional transition distributions for Alanine dipeptide starting from different metastable states. The starting distribution are sampled from the empirical distribution in the yellow region around the red point. (a) Distribution sampled from the MD simulation. (b) Distribution generated by the DeepGenMSM.