[Reviews · NeurIPS 2018]

Reviewer 1



In this paper, a deep generative Markov state model is proposed for inference of dynamical systems and prediction of trajectories. An encoder maps observed data to latent variables and the dynamics of the latent variables are modeled as Markov chains. Accordingly, a generator is trained to generate synthetic data from latent variables, and the transition matrix is estimated by the rewiring trick. The proposed method shows some potentials to the tasks of molecule dynamics. My main concerns are 1. Besides MSM, authors should consider more existing sequential models as baselines. 2. The architectures of proposed encoder and generator should be explained with details. The influence of the architectures on learning results should be analyzed in the experimental section. 3. The rationality and the superiority of using conditional energy distance (ED) should be explained with more details. To measure the distance between two distributions, there are many choices, e.g., KL divergence and Wasserstein distance. Compared with those metrics, what is the advantage of ED? Additionally, have authors tried to use other metrics to learn the proposed model? ---------------------------------- After rebuttal ---------------------------------- Authors solved my main concern about the rationality of ED, so I change my score to 7. Additionally, I still wish that authors can consider more baselines containing transition matrices (maybe not Markovian but interpretable transition matrices), e.g., nonnegative auto-regressive model. Overall, it is an interesting work.

Reviewer 2



This paper proposes a novel learning frame-work for Markov State Models of real valued vectors. This model can handle metastable processes i.e. processes that evolve locally in short time-scales but switch between a few clusters after very long periods. The proposed framework is based on a nice idea to decompose the transition from x1 to x2 to the probability that x1 belongs to a long-lived state and a distribution of x2 given the state. The first conditional probability is modeled using a decoding deep network whereas the second one can be represented either using a network that assigns weights to x2 or using a generative neural network. This is a very interesting manuscript. The proposed decomposition is a neat idea that allows to harness flexible representations to Markov processes with a complex structure. The model has the merit that it can efficiently simulate transitions from long-lived states that would take an extremely substantial number of steps using conventional simulations. Additionally, it allows to learn the stationary distribution of the process. The general flow of the paper is good, but some rearrangement and modifications could have make it easier to follow: (1) Mention that \tau is fixed, and in Equation 5 state that the approximation holds for sufficiently small \tau (2) The ‘rewiring trick’ is introduced after Equation 2 and in Figure 1, however it is not clear what it does. As far as I understand, the idea is to sample a transition from state $i$ to state $j$ by generating a sample $x$ from the generator of $j$ and passing this sample through the decoder. Then $K$ can be approximated using many sample. This trick also allows to estimate $K$ using the maximum likelihood approach as introduced in Equation 9. I think that explicit explanation after Equation 2, will make it easier to understand. (3) In Equation 1, y represents a state in the configuration space, whereas in Figure 1 it seems to represent an index of the latent space, which is confusing. (4) In the left-hand-side of Equation 11, x and y represent random vectors. It is not clear what do they represent in the right-hand-side. Are there averages of these variables? (5) In Equation 12, specify if the second line is conditioned on x_t. *** After rebuttal *** I have read the other reviews as well as the author's response. They address the issues I raised, which were oriented to improve the clarity of the paper, and therefore I maintain the original high score.

Reviewer 3



The manuscript introduces a new technique for learning Markov State Models from molecular trajectory data. The model consists of several novel ideas, including a generative component that makes it possible to reconstruct molecular structures from the latent space. Quality: The methodological contributions are convincing, and backed up by both source code and detailed derivations in supporting material (and references to relevant literature). Although the experiments focus on systems of modest size, and are thus mostly proof-of-concept, the modelling strategy itself has several non-standard components, which in my view makes the manuscript a significant Machine Learning contribution. The authors are honest about the strenths and weaknesses of their work. Clarity: The manuscript is of high quality. It is well written, and the text is accompanied with helpful figures. My only minor concern regarding clarity is that some of the subfigures in Fig 2 - 4 are difficult to read due to their small size. Originality: The presented methods are new, and represent a clear deviation from earlier work in this area. Significance: The authors present a convincing case that their approach has fundamental advantages over current techniques for estimating Markov State Models. It is thus likely to have considerable impact on the MSM field. The modelling strategy itself should be of interest more broadly in the Machine Learning community. Suggestions for corrections: In the text following eq (1), perhaps you should more explicitly introduce the hyperparameter m as the number of metastable states, to make it clearer to readers not familiar with the MSM literature. The discussion regarding the rewiring trick is a little brief. Perhaps you could explicitly write out the approximation either in the text or in supporting material. It's also not quite clear how many samples you draw in this approximation. Eq 7 is not quite clear to me. You write that \rho(y) is the empirical distribution, but how should I think of an empirical distribution over a continuous space y? Or are we talking about a distribution over the fixed set of input structures (in which case I would assume they are all equally likely - i.e. the exact same structure is never visited twice). Please elaborate. I found your Energy Distance procedure interesting, but could you perhaps briefly state why you would choose this over e.g. a Kullback-Leibler based approach? (since you use the Kullback-Leibler divergence later for measuring the difference between distributions (fig 2d). The dialanine system seems to be of rather modest size compared to the systems typically studied with MD. It would be nice if the paper concluded with some some statements on the extent to which the DeepResampleMSM and DeepGenMSM scale to such systems. Minor corrections: Line 42: "is conserves" -> "conserves" Line 130: "an generated trajectory" -> "any generated trajectory"? Update after rebuttal period: I only raised minor issues in my review, which the authors have addressed in their rebuttal. I therefore stand by my original (high) rating of the manuscript.